# Effect of lidocaine on salicylate-induced tinnitus in guinea pigs: A focus on the auditory cortex

Mutsumi Kenmochi, Kentaro Ochi *, Hirotsugu Kinoshita, Shigeru Kasugai, Manabu Nakamura, Manabu Komori

Department of Otolaryngology St. Marianna University School of Medicine, Kawasaki, Japan

☯ These authors contributed equally to this work.
* k2ochi@w6.dion.ne.jp

**Data Availability Statement:** All relevant data are within the manuscript and its Supporting information files.

## Abstract

This study aimed to investigate the effects of the intravenous administration of lidocaine in the auditory cortex after the systemic administration of salicylate. Healthy male albino Hartley guinea pigs were divided into two groups. The control group received only lidocaine, whereas the experimental group received lidocaine after checking for the effects of salicylate. Extracellular recordings of spikes in the primary auditory cortex and dorsocaudal areas in healthy albino Hartley guinea pigs were continuously documented (pre- and post-lidocaine, pre- and post-salicylate, and post-salicylate after adding lidocaine to post-salicylate). We recorded 160 single units in the primary auditory cortex from five guinea pigs and 155 single units in the dorsocaudal area from another five guinea pigs to confirm the effects of lidocaine on untreated animals. No significant change was detected in either the threshold or $Q_{10dB}$ value after lidocaine administration in the primary auditory cortex and dorsocaudal areas. Spontaneous firing activity significantly decreased after lidocaine administration in the primary auditory cortex and dorsocaudal areas. Next, we recorded 160 single units in the primary auditory cortex from five guinea pigs and 137 single units in the dorsocaudal area from another five guinea pigs to determine the effects of lidocaine on salicylate-treated animals. The threshold was significantly elevated after salicylate administration; however, no additional change was detected after adding lidocaine to the primary auditory cortex and dorsocaudal areas. Regarding the $Q_{10dB}$ value, lidocaine negated the significant changes induced by salicylate in the primary auditory cortex and dorsocaudal areas. Moreover, lidocaine negated the significant changes in spontaneous firing activities induced by salicylate in the primary auditory cortex and dorsocaudal areas. In conclusion, changes in the $Q_{10dB}$ value and spontaneous firing activities induced by salicylate administration are abolished by lidocaine administration, suggesting that these changes are related to the presence of tinnitus.

**Funding:** The author(s) received no specific funding for this work.

**Competing interests:** The authors have declared that no competing interests exist.

## Introduction

Lidocaine was first synthesized in 1943 by Swedish chemist Nils Löfgren [1]. It was later introduced for medical use in 1948. Local anesthetics are classified into two groups: amine and ester groups [2]. フォームの始まり Lidocaine is classified as an amino group, different from the ester group, such as procaine, that had been used until 1948. Lidocaine has been widely used because its anesthetic potency is twice that of procaine and it produces a greater anesthetic effect in a deeper and wider area. In addition to its local anesthetic effect, the suppressive effect of procaine on tinnitus was reported in 1936 [3]. The suppressive effect of lidocaine on tinnitus was first reported in 1961 [4]. Thereafter, several clinical studies have demonstrated the effectiveness of lidocaine on tinnitus [5–9]. Initially, the method of lidocaine administration was mainly intravenous [5–8]; however, recently, percutaneous administration has also been applied for easy intake [9]. However, the efficacy rates have varied, and although it is effective in some cases, problems such as continuity remain.

These administration methods are based on the systemic administration of lidocaine, although administration to the ear has also been reported [10–12]. When administered through the ear, the target organ or the site of the first action of the drug is limited to the periphery of the ear (sensory receptors of the neural elements in the cochlea or auditory nerve) and not the center of the auditory conduction pathway, potentially modulating the transmission of tinnitus-related signals to the center of the auditory conduction pathway. This would only be effective if the cause of tinnitus was limited to the area around the ear. From this perspective, Coles et al. [12] concluded that this form of treatment was insufficiently effective to offset its low acceptability.

Salicylate causes transient hearing loss and tinnitus in humans when ingested in large doses. Similar effects have been observed in animals. Therefore, several animal studies have been conducted on salicylate to investigate the causes of hearing loss and tinnitus. Recently, we reported on neuronal activity in the auditory cortex of guinea pigs before and after salicylate administration [13]. We observed that salicylate caused a significant threshold elevation, and the $Q_{10dB}$ value changed differently changed depending on the auditory area of the guinea pigs: an increase in the primary auditory cortex (AI) and a decrease in the dorsocaudal area (DC). Additionally, salicylate induced a change in single-unit spontaneous firing activity in the auditory cortex of guinea pigs, with a decrease in AI and an increase in DC. We hypothesized that these changes in the spontaneous firing activity in the auditory cortex could be a neural element of tinnitus.

In this study, we confirmed the effects of lidocaine on the auditory cortex of guinea pigs. Subsequently, salicylate was administered in the same setting as that in a previous study [13], and after confirming the effect, lidocaine was intravenously administered just after the second session (approximately after 60–90 min after salicylate administration) when the effect was stable and its effect was examined.

## Materials and methods

### Experiment 1

We conducted two types of experiments. Ten animals were allotted for the first experiment to examine the effect of lidocaine on untreated animals. Ten animals were used in the second experiment to evaluate the effect of lidocaine on animals treated with the systemic administration of salicylate.

In the first experiment, control data were collected in the same manner as in a previous study [13]. After obtaining the baseline (control) data (before the lidocaine session), a small skin incision was made around the right hip joint to expose the femoral vein for the first experiment. An intravenous injection of lidocaine (1 mg/kg) was slowly administered through the

right femoral vein. Approximately 10 min after the lidocaine injection, both stimulated and spontaneous activities were recorded for lidocaine data (after the lidocaine session).

## Experiment 2

The second experiment was conducted using the same method as previously described, except for one point [13]. The only difference was the addition of an intravenous lidocaine injection after the second session. After collecting baseline (control) data, 200 mg/kg (30–60 mg/mL) sodium salicylate was slowly administered intraperitoneally. Approximately 30 min after the administration of salicylate, both stimulated and spontaneous activities were recorded for post-salicylate data. The recording was continued twice at serial intervals after salicylate administration (first session, approximately 30–60 min; second session, approximately 60–90 min). After the second session, an intravenous injection of 1 mg/kg lidocaine was slowly administered through the right femoral vein. Approximately 10 min after lidocaine injection, both stimulated and spontaneous activities were recorded as lidocaine data (after the lidocaine session, approximately 100–130 min).

## Subjects and surgical procedure

Because the surgical technique was the same as that in a previous study [13], it is briefly described here. The experiments were conducted on 20 healthy albino Hartley guinea pigs weighing 340–690 g (mean±1 standard error of measurement, 484±25 g).

Guinea pigs were subcutaneously injected with 0.1 mL of atropine sulfate (0.6 mg/mL), followed by an intraperitoneal injection of sodium pentobarbital (32.5–65 mg/mL) at a dose of 25 mg/kg bodyweight. After a 15-min interval, local anesthesia around the trachea was induced by subcutaneously injecting Xylocaine® (a mixture of lidocaine hydrochloride and epinephrine), and a tracheostomy was performed. General anesthesia was maintained through tracheostomy via artificial ventilation using a mixture of 2% sevoflurane and 98% room air. Anesthesia depth was monitored based on heart rate and response to body pinch at 30-min intervals, with the sevoflurane dosage adjusted in 2% increments when necessary. Body temperature was monitored and maintained at approximately 37±1°C using a thermostatically controlled unit. After achieving stable anesthesia, the guinea pigs were paralyzed with an intramuscular injection of Relaxin® (suxamethonium chloride) at a dose of 50 mg/kg body weight. Subsequently, the guinea pigs were placed in a double-walled sound-attenuating room on a vibration isolation frame. After shaving the heads of the guinea pigs, Xylocaine® was subcutaneously injected into the right temporal region before incising the skin overlying the skull.

The skin flap was removed, and the overlying connective tissue was cleared from the skull. Local landmarks on the skull, such as the lateral and coronal sutures, were verified. An 8-mm-diameter hole was drilled along the lateral suture, which was occasionally enlarged with small bone rongeurs to expose the pseudosylvian sulcus for complete exposure of the AI and DC. The dura was opened, and the brain was covered with light mineral oil, adding more as required.

At the end of the experiments, the guinea pigs were sacrificed with an overdose of sodium pentobarbital. The guinea pigs were maintained and used in accordance with protocols (#17591807) approved by the Life and Environmental Science Animal Care Committee of the St. Marianna University School of Medicine in Kawasaki, Japan.

## Acoustic stimulus presentation

Acoustic stimuli were presented using a loudspeaker (JBL-2450h, JBL Inc., Northridge, CA) placed 50 cm in front of the head of the guinea pig. Calibration and monitoring were

performed using a condenser microphone (NA-41; Rion Ltd., Tokyo, Japan) positioned above the animal's head. The stimuli were generated and transferred to the DSP boards of the TDT-2 sound delivery system (Tucker-Davis Technologies, Alachua, FL). The tone pip ensemble consisted of three sequences of 27 tone bursts covering five octaves (500–17152 Hz) in a pseudo-random order at a fixed intensity. Characteristic frequencies (CFs) were obtained from the neural response frequency to the lowest stimulus intensity, and thresholds were determined by the visual detection of responses to the lowest stimulation in CFs. The $Q_{10dB}$ value represents a CF-to-bandwidth ratio of 10 dB above the CF, indicating relative frequency tuning sharpness.

## Recordings from the auditory cortex

Extracellular spike recording was performed using the Michigan probe (A4×4-10mm-100-200-413-A16, NeuroNexus, Ann Arbor, MI), each consisting of four shanks with four microelectrode sites, totaling 16 microelectrode sites (surface area, 413 $\mu m^2$) with impedances between 1 and 2 MΩ. High-impedance head stages (RA16AC; Tucker-Davis Technologies) were used. The microelectrodes were organized in a 4×4 configuration, with 100-μm spacing between individual microelectrodes and approximately 200 μm between adjacent shanks. Initially, all 16 microelectrodes were utilized for recording; however, because of occasional recording difficulties, a subsequent selection involved the use of two terminal microelectrodes from each shank, resulting in a total of eight microelectrode sites.

Recordings commenced immediately caudal to the pseudosylvian sulcus at a point known for low-frequency units in the AI area [14]. The recording probe was then caudally moved to locations with higher-frequency units in the AI. The border between the AI and DC was identified using a line connecting the points with a tonotopic gradient reversal. Subsequently, the recording probe was moved beyond this border to sites with lower frequency units in the DC. The recording targets included both AI and DC, with 10 animals selected for AI recordings and the remaining 10 animals for DC recordings. Reference and ground electrodes were positioned on the neck muscles. The probe arrays were nearly orthogonally oriented towards the cortical surface and manually advanced using a motorized four-axis micromanipulator (MX7600R; Siskiyou, Grants Pass, OR). Recording depth ranged between 600 and 900 μm, likely placing electrodes in the deep layer III or IV.

The recorded potentials were amplified $10^4$ times using a pair of amplifiers (RA 16P; Tucker-Davis Technologies) and processed using a multichannel data acquisition system (RA 16; Tucker-Davis Technologies). Neural activity was recorded using the Brainware software (Tucker-Davis Technologies). The neural data were digitized at 12 kHz and bandpass filtered from 300 to 3000 Hz at 9 dB/octave. Spikes were identified online using the trigger levels above the noise level (usually ±14.28 μV). Artifact rejection was used to extract the action potential candidates from the digitized input. Without artifact rejection, any voltage peaks exceeding either trigger level were recorded. With artifact rejection, only the candidates for biphasic action potentials with both peaks exceeding the respective noise level settings were recorded.

Details of the spike separation method are presented in a previous study [13]. The action potentials with peaks exceeding the respective trigger levels were recorded. Each spike was classified by selecting clusters formed by selecting the x- and y-axes among the features measured by the Brainware software (Peak 1, Peak 2, Peak to Peak, and Trig to Trig). We usually select Peak to Peak, or Peak 1 on the x-axis and Peak 2 on the y-axis. We continuously recorded on the same single unit throughout the experimental session. We confirmed the waveforms to check the single-unit separation by selecting clusters.

(A) (B)

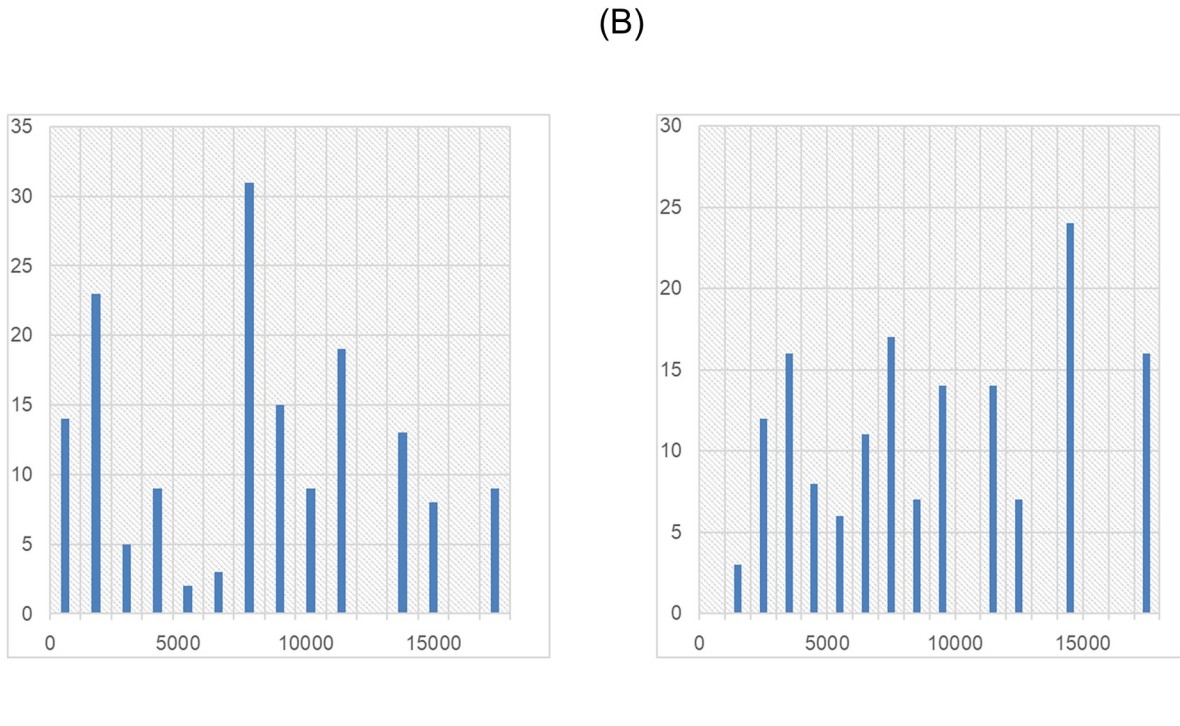

Characteristic frequency（Hz） Characteristic frequency（Hz）

**Fig 1. Distribution of Characteristic Frequencies (CFs).** The distributions of the CFs of the primary auditory cortex (AI) and dorsocaudal area (DC) are separately shown in (A) AI and (B) DC.

We typically recorded four units per electrode site. Spontaneous activity was based on recordings of 15-min duration without sound, whereas stimulated activity was based on recordings of 15-min duration with sound. It generally took approximately 120 min from the initiation of recording to achieve a stable recording and confirm the recording areas (AI or DC). After confirming stable recordings, stimulated and spontaneous activities were recorded as baseline (control) data.

## Statistical analyses

All statistical tests were performed using Bell Curve for Excel (version 4.06) (Social Survey Research Information Co., Tokyo, Japan). A p-value of <0.05 was considered statistically significant. Statistical tests for the threshold, CF, $Q_{10dB}$ value, and firing rate between the AI and DC were based on the Mann–Whitney U test. Data obtained before and after lidocaine administration were analyzed using the Wilcoxon signed-rank test. Data obtained before and after the administration of salicylate and the addition of lidocaine were analyzed using the Friedman test. Scheffé's paired comparison test was used to compare the two groups using the Friedman test.

## Results

### Experiment 1

We recorded 160 single units in the AI in five animals and 155 single units in the DC in another five animals to evaluate the effect of lidocaine on untreated guinea pigs. We were not able to continuously record the five single units of the DC owing to burst firing. Fig 1 shows

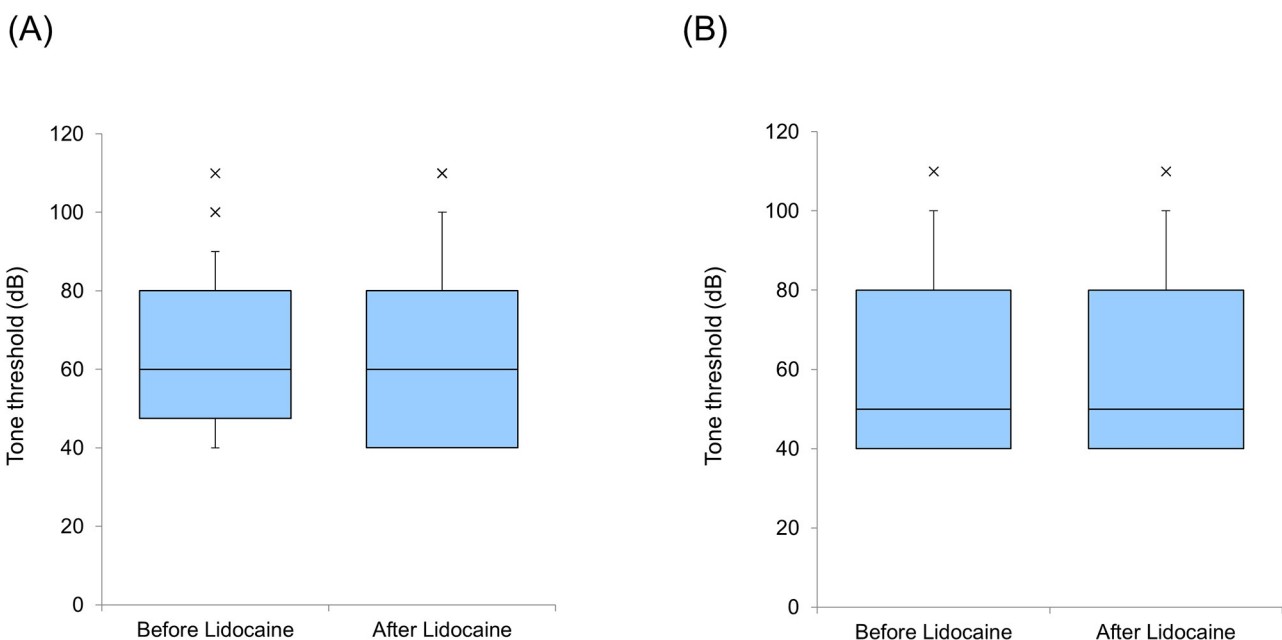

**Fig 2. Thresholds Before and After Lidocaine Administration (A: Primary auditory cortex [AI], B: Dorsocaudal area [DC]).** Each horizontal line indicates the tenth, first quartile, median, third quartile, and 90th percentiles.

the distribution of the CFs of the AI and DC. The CFs of the AI were significantly lower than those of the DC (p<0.001, Mann–Whitney U test).

No significant difference was observed in the threshold change in the AI (*p* = 0.49, Wilcoxon signed-rank test) and DC (p = 0.52, Wilcoxon signed-rank test) (Fig 2). No significant difference was noted between the thresholds of the AI and DC (p = 0.20, Mann–Whitney U test).

The $Q_{10dB}$ values of the AI and DC before and after lidocaine administration are shown in Fig 3 (A: AI, B: DC). No significant difference was observed in the changes in the $Q_{10dB}$ values in the AI (p = 0.28, Wilcoxon signed-rank test) and DC (p = 0.87, Wilcoxon signed-rank test). The $Q_{10dB}$ value before the session in the AI was significantly smaller than that before the session in the DC (p<0.001, Mann–Whitney U test).

The spontaneous firing activity of both the AI and DC significantly decreased after lidocaine administration (AI and DC, p<0.001, Wilcoxon signed-rank test) (Fig 4). Spontaneous activity before the session in the AI was significantly lower than that before the session in the DC (p<0.05, Mann–Whitney U test).

### Experiment 2

We recorded 160 single units in the AI in five animals and 154 single units in the DC in another five animals to evaluate the effect of lidocaine on salicylate-treated guinea pigs. We were not able to record continuously in the six single units of the DC owing to burst firing. Fig 5 shows the distribution of the CFs of the AI and DC. The CFs of the AI were significantly higher than those of the DC (p<0.001, Mann–Whitney U test).

The thresholds of the AI and DC before, in the first, in the second, and after the lidocaine session are separately shown in Fig 6 (A: AI, B: DC). A significant difference (p<0.001) was observed in the thresholds of both the AI and DC (Friedman test).

(A)                                              (B)

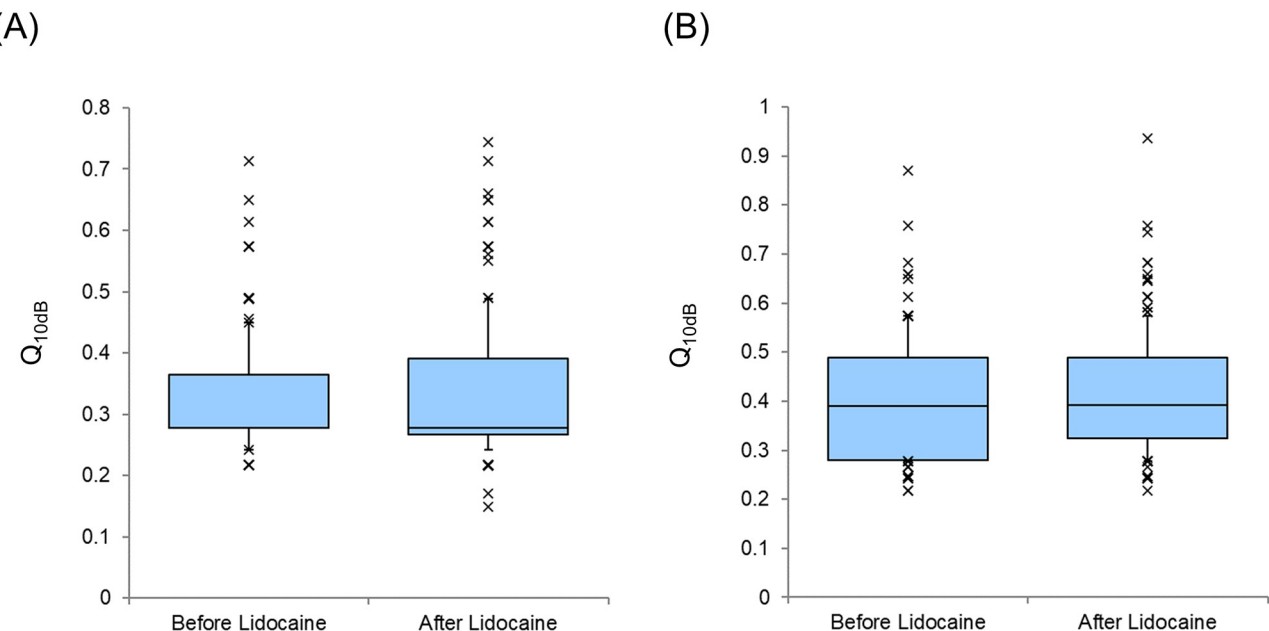

**Fig 3. $Q_{10dB}$ Values Before and After Lidocaine Administration (A: Primary auditory cortex, B: Dorsocaudal area).** Each horizontal line indicates the tenth, first quartile, median, third quartile, and 90th percentiles. Outliers are also plotted.

The AI thresholds in the first session were significantly higher than those before the session ($p < 0.05$, Scheffé's paired comparison). The AI thresholds in the second session were significantly higher than those before the session and in the first session ($p < 0.001$, Scheffé's paired comparison). The AI thresholds after the lidocaine session were significantly higher than those

(A)                                              (B)

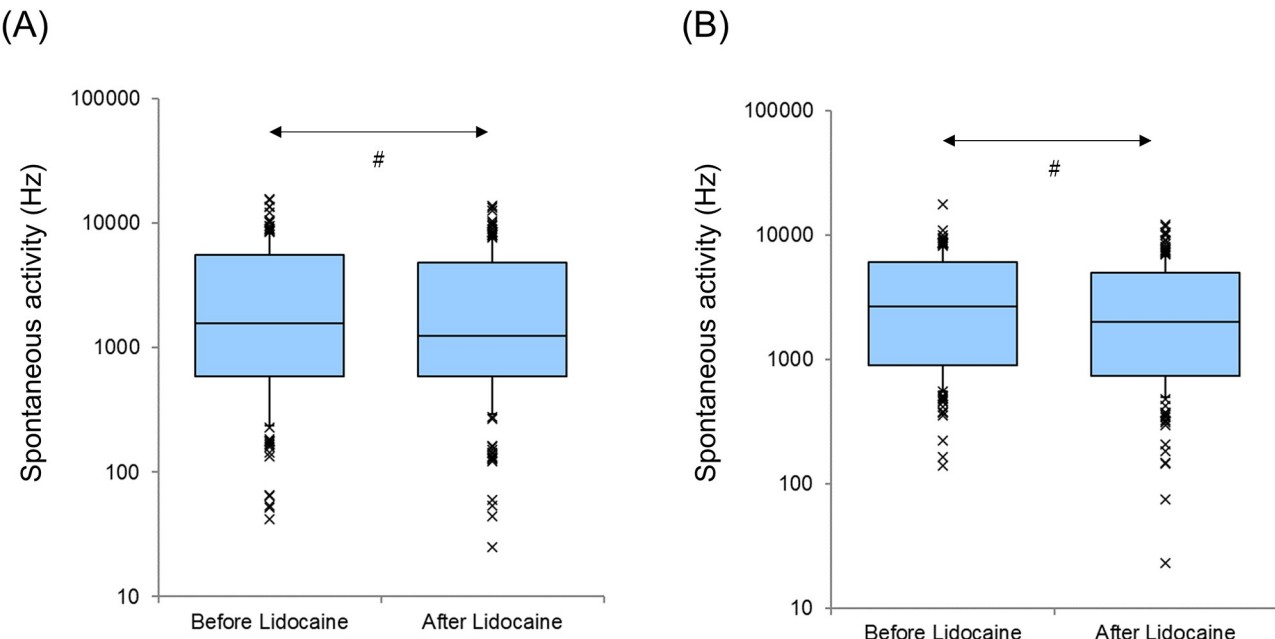

**Fig 4. Spontaneous Activity Before and After Lidocaine Administration (A: Primary auditory cortex, B: Dorsocaudal area).** A logarithmic scale is used. Each horizontal line indicates the tenth, first quartile, median, third quartile, and 90th percentiles. Outliers are also plotted. [#]$p < 0.001$.

(A) (B)

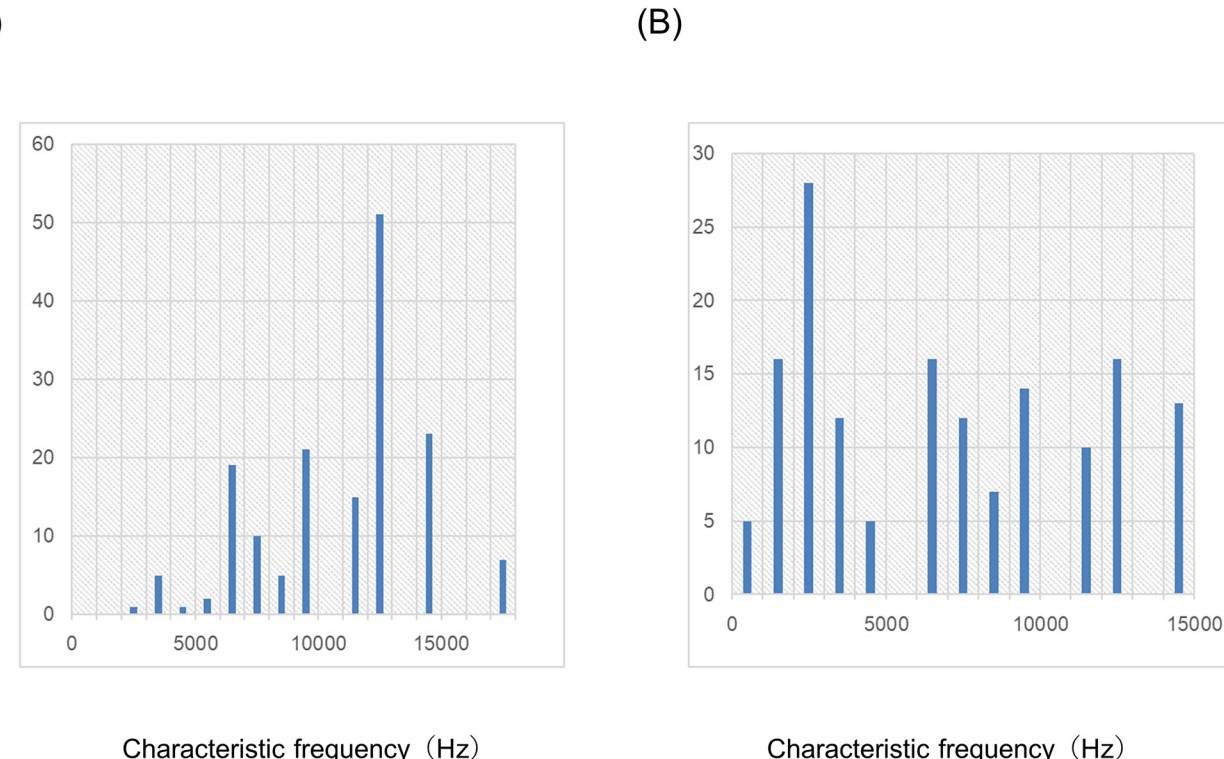

Characteristic frequency（Hz）　　　　　　Characteristic frequency（Hz）

**Fig 5. Distribution of Characteristic Frequencies (CFs).** The distributions of the CFs of the primary auditory cortex (AI) and dorsocaudal area (DC) are separately shown in (A) AI and (B) DC.

before the session and in the first session (p<0.001, Scheffé's paired comparison). No significant difference was noted between the thresholds of the AI in the second session and those after the lidocaine session (p = 0.92, Scheffé's paired comparison).

The DC thresholds in the first session were significantly higher than those before the session (p<0.001, Scheffé's paired comparison). The DC thresholds in the second session were significantly higher than those before the session (p<0.001, Scheffé's paired comparison). The AI thresholds after the lidocaine session were significantly higher than those before session and in the first session (p<0.001, Scheffé's paired comparison). No significant difference was observed in the DC thresholds between the first and second sessions (p = 0.23) and between the second session and after the lidocaine session (p = 0.80, Scheffé's paired comparison).

The thresholds before the session in the AI were significantly (p<0.001) lower than those in the DC (Mann–Whitney U test).

The $Q_{10dB}$ values of the AI and DC before, in the first, in the second, and after the lidocaine session are illustrated in Fig 7 (A: AI, B: DC). A significant difference ($p<0.001$) was detected in the $Q_{10dB}$ values of both the AI and DC (Friedman test). The $Q_{10dB}$ value in the AI of the second session was significantly higher than that before session and in the first session (p<0.001, Scheffé's paired comparison). The $Q_{10dB}$ value in the AI after the lidocaine session was significantly lower than that in the second session (p<0.001, Scheffé's paired comparison). No significant difference was observed between the $Q_{10dB}$ value in the AI before the session and in the first session ($p = 0.55$), before the session and after the lidocaine session (p = 0.85), or the first session and after the lidocaine session (p = 0.14, Scheffé's paired comparison).

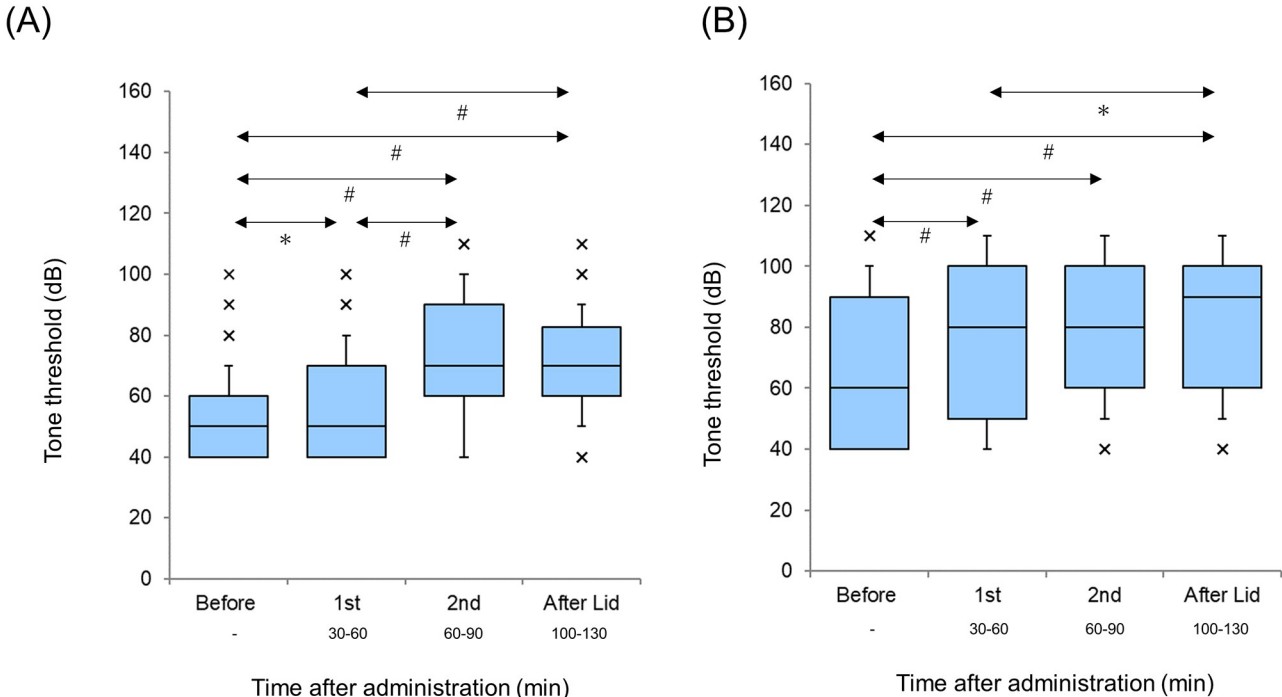

**Fig 6. Threshold at each session.** The thresholds for each session of the primary auditory cortex (AI) and dorsocaudal area (DC) are separately shown in (A) AI and (B) DC. Each horizontal line indicates the tenth, first quartile, median, third quartile, and 90th percentiles. Outliers are also plotted. (A) AI. (B) DC. *$p<0.05$, #$p<0.001$.

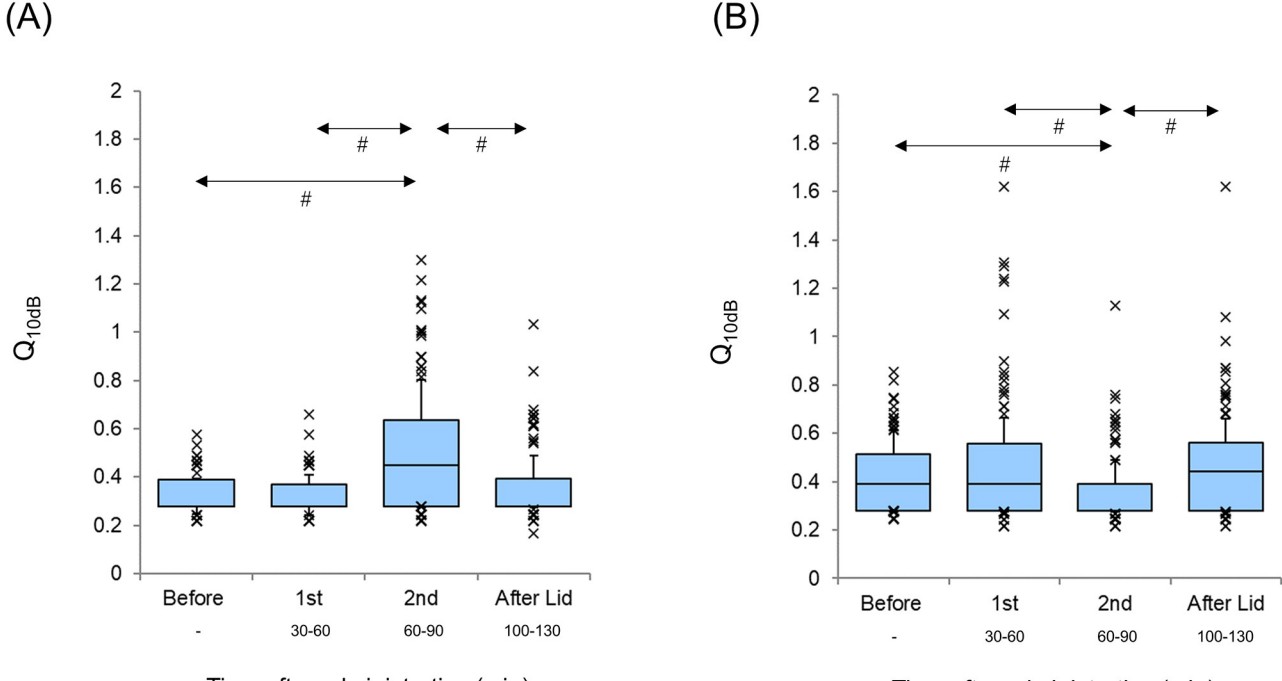

**Fig 7. $Q_{10dB}$ values at each session.** The $Q_{10dB}$ values at each session of the primary auditory cortex (AI) and dorsocaudal area (DC) are separately shown in (A) AI and (B) DC. Each horizontal line is the same as that in Fig 2. Outliers are also plotted. (A) AI. (B) DC. #$p < 0.001$.

(A)

(B)

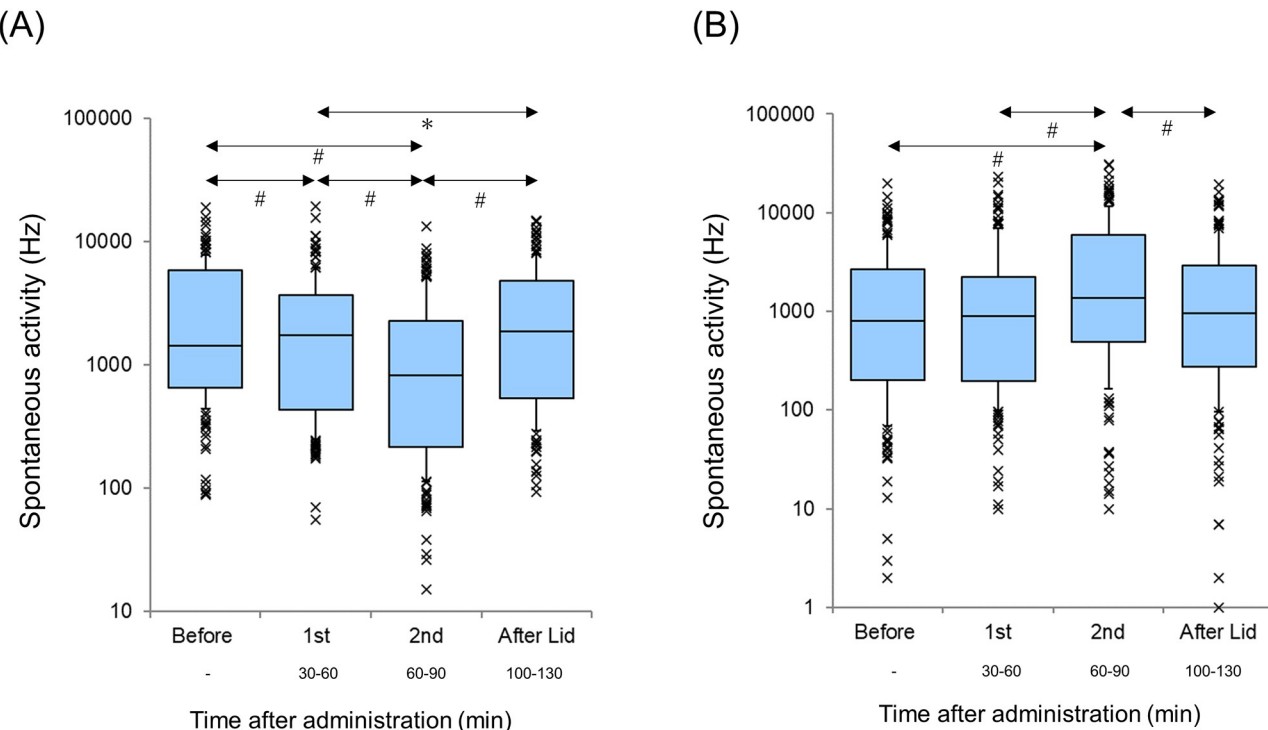

**Fig 8. Spontaneous firing rate at each session.** The spontaneous firing rates in each session of the primary auditory cortex (AI) and dorsocaudal area (DC) are separately shown in (A) AI and (B) DC. A logarithmic scale is used. Each horizontal line is the same as that in Fig 2. Outliers are also plotted. *p<0.05, #p<0.001.

The $Q_{10dB}$ value in the DC of the second session was significantly lower than that before the session (p<0.001, Scheffé's paired comparison). Furthermore, the $Q_{10dB}$ value in the DC of after the lidocaine session was significantly higher than that of the second session (p<0.001, Scheffé's paired comparison). No significant difference was observed between the $Q_{10dB}$ value in the DC before the session and the first session (p = 0.95), that before the session and after the lidocaine session (p = 0.99), or the first session and after the lidocaine session (p = 0.93, Scheffé's paired comparison). The $Q_{10dB}$ value before the session in the AI was significantly smaller than that before the session in the DC (p<0.001, Mann–Whitney U test).

Fig 8 shows the changes in the spontaneous firing rates in the AI and DC. A significant difference was observed in the spontaneous firing activity of both the AI and DC (p<0.001, Friedman test).

The spontaneous activities in the AI of the first and second sessions were significantly lower than those before the sessions (p<0.001, Scheffé's paired comparison). The spontaneous firing activities in the AI of the second session were significantly lower than those in the first session (p<0.001, Scheffé's paired comparison). The spontaneous firing activities in the AI after the lidocaine session were significantly higher than those of the first and second sessions (vs. the first session: p<0.05, vs. the second session: p<0.001, Scheffé's paired comparison). No significant difference was noted in the spontaneous activity in the AI before the session and after the lidocaine sessions (p = 0.54, Scheffé's paired comparison).

The spontaneous activity in the DC of the second session was significantly higher than that before the session and that of the first session (p<0.001, Scheffé's paired comparison). The spontaneous activities in the DC after the lidocaine session were significantly lower than those

of the second session (p<0.001, Scheffé's paired comparison). No significant difference was observed between the spontaneous activity in the DC before the session and the first session (p = 0.97), before the session and after the lidocaine session (p = 0.97), or the first session and after the lidocaine session (p = 0.80, Scheffé's paired comparison). The spontaneous firing before the session in the AI was significantly higher than that before the session in the DC (p<0.001, Mann–Whitney U test).

## Discussion

We recently confirmed the effect of the systemic administration of salicylate in 10 guinea pigs, and cortical changes could be related to the generation of tinnitus [13]. In the present study, we intravenously administered lidocaine, which suppresses tinnitus, to guinea pigs after the systemic administration of salicylate. First, the guinea pigs were treated with lidocaine, and its effects were examined; the threshold and $Q_{10dB}$ values were not changed, although spontaneous activity significantly decreased in the AI and DC. Yu and Chen [2] observed that lidocaine significantly suppressed the current-evoked firing of action potentials and suggested that lidocaine at low concentrations (30–100 μmol/L, clinically relevant concentration) reduced the excitability of primary cultured neurons of the inferior colliculus to suppress tinnitus, probably through the inhibition of voltage-gated sodium channels. The significant reduction in spontaneous activity in the AI and DC observed in the present study may also be due to the suppression of voltage-gated sodium channels, as they have shown.

We observed significant differences between the AI and DC groups (CFs, threshold, $Q_{10dB}$ value, and spontaneous activity). However, we recorded continuous cortical activity in a single animal during each experiment, which reduced the effects of individual differences between groups and allowed for a more reliable assessment of changes. Some studies examined the insertion of chronic electrodes [15]. In this case, the effects of anesthesia, the experimental environment, and other factors can be further reduced. Additional factors that may affect recordings from the auditory cortex include laboratory environment and animal condition. For the recording environment, the temperature in the laboratory was kept constant, and recording was performed in a soundproof room. Regarding animal condition, healthy animals were selected, body temperature was kept constant with a thermostatically controlled unit, pain was monitored based on heart rate and response to body pinch at 30-min intervals, and anesthesia depth was kept constant.

The thresholds altered by salicylate administration were not altered by additional lidocaine administration. The $Q_{10dB}$ values of the AI, which were increased by the administration of salicylate, were decreased by the additional administration of lidocaine, and the $Q_{10dB}$ values of the DC, which were decreased by the administration of salicylate, were increased by the additional administration of lidocaine. Both results were no longer significant compared with those before salicylate administration, indicating normalization by lidocaine. Salicylate significantly decreased the number of AI spontaneous firing and significantly increased the number of DC spontaneous firing, as previously reported [13]; however, the addition of lidocaine increased the number of AI spontaneous firing and decreased the number of DC spontaneous firing, showing no significant change from that before salicylate administration. Thus, the effects of salicylate were modified by lidocaine administration. In normal guinea pigs, lidocaine significantly decreased the number of spontaneous activities of both AIs and DCs. Therefore, it is easy to understand that lidocaine reduced the number of salicylate-induced increased DCs, although the increased number of salicylate-induced decreased AIs must be considered an effect of lidocaine on salicylate.

The salicylate-induced changes in the $Q_{10dB}$ values and spontaneous firing activity were corrected by the additional administration of lidocaine, whereas the salicylate-induced threshold changes were not corrected by the additional administration of lidocaine, presumably because the salicylate-induced threshold changes occurred at different locations than the $Q_{10dB}$ values and spontaneous firing activity. Chen et al. [16] suggested that hearing loss induced by systemic salicylate treatment was cochlear in origin, whereas sound-evoked hyperactivity observed in the auditory cortex, medial geniculate body, and lateral amygdala originated in the central nervous system.

The $Q_{10dB}$ values and spontaneous firing activity were altered by salicylate but were modified to the prior salicylate administration level by the additional administration of lidocaine. This suggests that changes in the $Q_{10dB}$ values and spontaneous firing activity are associated with tinnitus. The mechanism by which lidocaine alleviates salicylate-induced changes in $Q_{10dB}$ values and spontaneous firing may involve ion channels and neurotransmitter systems within the auditory pathway.

Lidocaine works primarily by blocking voltage-gated sodium channels, thereby inhibiting the generation and propagation of action potentials in neurons [2]. In tinnitus, aberrant neural activity, possibly due to hyperexcitability, may contribute to sound perception. By blocking sodium channels, lidocaine can potentially reduce abnormal neural firing, leading to a decrease in tinnitus perception. Lidocaine can also modulate potassium channels, although its effects on these channels are less pronounced compared with those on sodium channels. Lidocaine may further regulate neuronal excitability and contribute to the suppression of tinnitus by affecting potassium channels; however, Yu and Chen [2] reported that lidocaine might not be involved in producing suppressive effects on tinnitus at a clinical dose.

Regarding the function of neurotransmitters, salicylate-induced tinnitus is associated with increased glutamate release in the auditory system [17]. Lidocaine reduces glutamate release by inhibiting presynaptic sodium channels, thereby mitigating the excitotoxicity and neuronal damage associated with tinnitus [2]. In addition, lidocaine enhances the activity of gamma-aminobutyric acid (GABA), a primary inhibitory neurotransmitter in the brain [2]. Dysfunction of the GABAergic system has been implicated in the pathology of tinnitus. By enhancing GABAergic inhibition, lidocaine counteracts the hyperexcitability of auditory neurons induced by salicylate, leading to the suppression of tinnitus. However, glutamate, GABA, and transient outward potassium channels have no inhibitory effects on tinnitus at clinical doses of lidocaine [2].

By targeting both ion channels and the neurotransmitter system, lidocaine may have suppressed salicylate-induced tinnitus in animal experiments. Its ability to modulate neuronal excitability and restore balance within the auditory pathway highlights its therapeutic potential in the treatment of tinnitus. Understanding the precise mechanisms underlying the anti-tinnitus effects of lidocaine can provide valuable insights for the development of novel pharmacological interventions targeting tinnitus.

Tinnitus, the perception of sound without an external source, can be induced in animal models using salicylate. Animal studies have provided valuable insights into the mechanisms underlying tinnitus generation and potential therapeutic interventions. The neural correlates of tinnitus are mainly based on spontaneous neural activities such as spontaneous firing rates and pairwise spontaneous spike-firing correlations [18]. In fact, we previously reported that salicylate altered the spontaneous firing activity of the guinea pig auditory cortex [13] and confirmed that the altered spontaneous firing activity was recovered to pre-salicylate levels by additional lidocaine administration in the present study. To clinically apply these findings to the treatment of tinnitus in humans, it is necessary to further investigate tinnitus models other than salicylate and assess the use of drugs other than lidocaine that are effective against

tinnitus. It is crucial to ensure that animal model findings are carefully interpreted and validated in human clinical studies to ascertain their relevance and applicability to human tinnitus.

## Conclusion

Intravenous administration of lidocaine did not result in significant changes in threshold or $Q_{10dB}$ values but significantly decreased the spontaneous activity in the AI and DC. Systemic salicylate administration induced the same results as those observed in a previous study [13]. After the administration of salicylate (session 2), lidocaine was additionally administered intravenously. The thresholds did not significantly change, although both the $Q_{10dB}$ values and spontaneous activities returned to the level before the administration of salicylate. This is the first report confirming the normalization of salicylate-induced changes in the auditory cortex, the final point of sound perception in guinea pigs, by lidocaine, and the finding is significant as it can be applied to further studies.

## Supporting information

**S1 Dataset.**
(XLSX)

**S2 Dataset.**
(XLSX)

## Acknowledgments

The authors would like to thank Editage for their English language editing services.

## Author Contributions

**Data curation:** Mutsumi Kenmochi, Kentaro Ochi, Hirotsugu Kinoshita.

**Formal analysis:** Mutsumi Kenmochi, Kentaro Ochi, Hirotsugu Kinoshita.

**Investigation:** Mutsumi Kenmochi, Kentaro Ochi, Hirotsugu Kinoshita, Shigeru Kasugai.

**Methodology:** Mutsumi Kenmochi, Kentaro Ochi.

**Project administration:** Mutsumi Kenmochi, Kentaro Ochi, Manabu Nakamura.

**Supervision:** Kentaro Ochi, Manabu Komori.

**Writing – original draft:** Kentaro Ochi.

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
