## [Decision Letter · Decision Letter 0]

2 Jun 2024

PONE-D-24-12496Effect of lidocaine on salicylate-induced tinnitus in guinea pigs: a focus on the auditory cortexPLOS ONE

Dear Dr. Ochi,

Thank you for submitting your manuscript to PLOS ONE. After careful consideration, we feel that it has merit but does not fully meet PLOS ONE’s publication criteria as it currently stands. Therefore, we invite you to submit a revised version of the manuscript that addresses the points raised during the review process.

We look forward to receiving your revised manuscript.

Kind regards,

Trung Quang Nguyen

Academic Editor

PLOS ONE

Journal Requirements:

2. Please upload a new copy of Figures 1 - 8 as the detail is not clear. Please follow the link for more information: https://blogs.plos.org/plos/2019/06/looking-good-tips-for-creating-your-plos-figures-graphics/" https://blogs.plos.org/plos/2019/06/looking-good-tips-for-creating-your-plos-figures-graphics/

Reviewers' comments:

Reviewer's Responses to Questions

**Comments to the Author**

1. Is the manuscript technically sound, and do the data support the conclusions?

Reviewer #1: Yes

Reviewer #2: Yes

2. Has the statistical analysis been performed appropriately and rigorously? 

Reviewer #1: Yes

Reviewer #2: Yes

3. Have the authors made all data underlying the findings in their manuscript fully available?

Reviewer #1: Yes

Reviewer #2: Yes

4. Is the manuscript presented in an intelligible fashion and written in standard English?

Reviewer #1: Yes

Reviewer #2: Yes

5. Review Comments to the Author

**Reviewer #1: **After reviewing the paper "Effect of lidocaine on salicylate-induced tinnitus in guinea pigs: a focus on the auditory cortex," it is clear that the manuscript is based on compelling empirical evidence and makes a unique contribution. Only minor revisions are required before it can be published. The manuscript contains an elaborate old literature review that needs to be updated. The material and method must be divided into Experminat 1 and Experminat 2, as shown in the result section.

**Reviewer #2: **Dear authors,

I found your research quite intriguing and your explanation of these results is sufficient. However, please edit your manuscript in accordance with the comments below.

Comments to the Author

Line 64. Please rewrite “tinnitus is limited" to be " tinnitus was limited ".

Line 72. Please rewrite " value differently changed " to be " value changed differently".

Line 82. Please remove a comma. “stable, and” to be “stable and”

Line 93. Please add an article " and tracheostomy “to be " and a tracheostomy “.

Line 100. Please replace " bodyweight " to be " body weight ".

Line 141. Please rewrite " higher frequency " to be " higher-frequency ".

Line 144. Please rewrite " both the AI and DC" to be " both AI and DC".

Line 146. Please replace " on neck muscles" with " on the neck muscles".

Line 147. Please rewrite " oriented toward" to be " oriented towards".

Line 165. Please add a comma " Peak or Peak 1 " to be " Peak, or Peak 1 ".

Line 179. Please remove comma " as that in “to be " as in “.

Line 187. Please rewrite " of intravenous" to be " of an intravenous”.

Lines 205. Please rewrite " and addition " to be " and the addition”.

Lines 219, 259, 284, 311, 340. Please rewrite " shown in the (A) AI and (B) DC." to be " shown in (A) AI and (B) DC.".

Line 362. Please add an article " experimental environment" to be " the experimental environment”.

6. PLOS authors have the option to publish the peer review history of their article (what does this mean?). If published, this will include your full peer review and any attached files.

Reviewer #1: No

Reviewer #2: No

---

## [Author Response · Author response to Decision Letter 0]

11 Jun 2024

Academic Editor

Dr. Trung Quang Nguyen

Thank you for reviewing our manuscript.

We carefully considered the comments provided by you and the other reviewer, and the manuscript was revised accordingly.

In addition, some explanations of the graphs were unclear and have been corrected.

1. We have verified that the manuscript meets the style requirements of PLOS ONE, including the requirements for file names.

2. We have uploaded a clearer file of Fig 1-8.

3. We have verified that the reference list is complete and correct.

Line 311. We rewrote“before and after" to be " before session and in ".

Line 336. We rewrote“before and after" to be " before session and in ".

Line 337. We rewrote " before and during " to be " before session and in ".

Line 338. We rewrote " than that after " to be " than that in ".

Line 340. We rewrote " before the first session and that after " to be " before the session and in ".

Line 340. We rewrote " that before the lidocaine session and that " to be " before the session and ".

Line 341. We rewrote " and that during the first session and that " to be "or the first session and ".

Line 344. We rewrote " before the first session " to be " before the session ".

Line 345. We inserted " the lidocaine session " to be " after the lidocaine session ".

Line 347. We rewrote " spontaneous activity " to be " Q10dB value ".

Line 347. We rewrote " before the first session and that after " to be " before the session and ".

Line 348. We rewrote " before the lidocaine session and that " to be " before the session and ".

Line 348. We rewrote " and that during the first session and that " to be " or the first session and ".

Line 387. We rewrote " before and after " to be " before the session and after ".

Line 390. We rewrote " before and during " to be " before the session and that of ".

Line 391 We rewrote " the lidocaine session " to be "after the lidocaine session ".

Line 394. We rewrote " before the first session and that " to be " before the session and ".

Line 394. We rewrote " that before the lidocaine session and that " to be " before the session and ".

Line 395. We rewrote " and that during the first session and that " to be "or the first session and ".

 

Reviewer #1

Thank you for reviewing our manuscript.

We carefully considered the comments provided by you, the other reviewer and academic editor, and the manuscript was revised accordingly.

In addition, some explanations of the graphs were unclear and have been corrected.

We have updated our elaborate old literature review.

We have divided the materials and methods into Experminute 1 and Experminute 2, as shown in the Results column.

Line 311. We rewrote“before and after" to be " before session and in ".

Line 336. We rewrote“before and after" to be " before session and in ".

Line 337. We rewrote " before and during " to be " before session and in ".

Line 338. We rewrote " than that after " to be " than that in ".

Line 340. We rewrote " before the first session and that after " to be " before the session and in ".

Line 340. We rewrote " that before the lidocaine session and that " to be " before the session and ".

Line 341. We rewrote " and that during the first session and that " to be "or the first session and ".

Line 344. We rewrote " before the first session " to be " before the session ".

Line 345. We inserted " the lidocaine session " to be " after the lidocaine session ".

Line 347. We rewrote " spontaneous activity " to be " Q10dB value ".

Line 347. We rewrote " before the first session and that after " to be " before the session and ".

Line 348. We rewrote " before the lidocaine session and that " to be " before the session and ".

Line 348. We rewrote " and that during the first session and that " to be " or the first session and ".

Line 387. We rewrote " before and after " to be " before the session and after ".

Line 390. We rewrote " before and during " to be " before the session and that of ".

Line 391 We rewrote " the lidocaine session " to be "after the lidocaine session ".

Line 394. We rewrote " before the first session and that " to be " before the session and ".

Line 394. We rewrote " that before the lidocaine session and that " to be " before the session and ".

Line 395. We rewrote " and that during the first session and that " to be "or the first session and ".

Reviewer #2

Thank you for reviewing our manuscript.

We carefully considered the comments provided by you, the other reviewer and academic editor, and the manuscript was revised accordingly.

In addition, some explanations of the graphs were unclear and have been corrected.

We have followed your suggestion and changed the sentence.

Line 64. We rewrote“tinnitus is limited" to be " tinnitus was limited ".

Line 72. We rewrote " value differently changed " to be " value changed differently".

Line 82. We removed a comma. “stable, and” to be “stable and”

Line 93. We added an article " and tracheostomy” “to be " and a tracheostomy “.

Line 100. We replaced " bodyweight " to be " body weight ".

Line 141. We rewrote " higher frequency " to be " higher-frequency ".

Line 144. We rewrote " both the AI and DC" to be " both AI and DC".

Line 146. We replaced " on neck muscles" with " on the neck muscles".

Line 147. We rewrote " oriented toward" to be " oriented towards".

Line 165. We added a comma " Peak or Peak 1 " to be " Peak, or Peak 1 ".

Line 179. We removed comma " as that in “to be " as in “.

Line 187. We rewrote " of intravenous" to be " of an intravenous”.

Lines 205. We rewrote " and addition " to be " and the addition”.

Lines 219, 259, 284, 311, 340. We rewrote " shown in the (A) AI and (B) DC." to be " shown in (A) AI and (B) DC.".

Line 362. We added an article " experimental environment" to be " the experimental environment”.

In addition, some explanations of the graphs were unclear and have been corrected.

Line 311. We rewrote“before and after" to be " before session and in ".

Line 336. We rewrote“before and after" to be " before session and in ".

Line 337. We rewrote " before and during " to be " before session and in ".

Line 338. We rewrote " than that after " to be " than that in ".

Line 340. We rewrote " before the first session and that after " to be " before the session and in ".

Line 340. We rewrote " that before the lidocaine session and that " to be " before the session and ".

Line 341. We rewrote " and that during the first session and that " to be "or the first session and ".

Line 344. We rewrote " before the first session " to be " before the session ".

Line 345. We inserted " the lidocaine session " to be " after the lidocaine session ".

Line 347. We rewrote " spontaneous activity " to be " Q10dB value ".

Line 347. We rewrote " before the first session and that after " to be " before the session and ".

Line 348. We rewrote " before the lidocaine session and that " to be " before the session and ".

Line 348. We rewrote " and that during the first session and that " to be " or the first session and ".

Line 387. We rewrote " before and after " to be " before the session and after ".

Line 390. We rewrote " before and during " to be " before the session and that of ".

Line 391 We rewrote " the lidocaine session " to be "after the lidocaine session ".

Line 394. We rewrote " before the first session and that " to be " before the session and ".

Line 394. We rewrote " that before the lidocaine session and that " to be " before the session and ".

Line 395. We rewrote " and that during the first session and that " to be "or the first session and ".

---

## [Decision Letter · Decision Letter 1]

21 Jun 2024

Effect of lidocaine on salicylate-induced tinnitus in guinea pigs: a focus on the auditory cortex

PONE-D-24-12496R1

Dear Dr. Kentaro Ochi,

We’re pleased to inform you that your manuscript has been judged scientifically suitable for publication and will be formally accepted for publication once it meets all outstanding technical requirements.

Kind regards,

Trung Quang Nguyen

Academic Editor

PLOS ONE

Additional Editor Comments (optional):

Reviewers' comments:

Reviewer's Responses to Questions

**Comments to the Author**

1. If the authors have adequately addressed your comments raised in a previous round of review and you feel that this manuscript is now acceptable for publication, you may indicate that here to bypass the “Comments to the Author” section, enter your conflict of interest statement in the “Confidential to Editor” section, and submit your "Accept" recommendation.

Reviewer #1: All comments have been addressed

Reviewer #2: All comments have been addressed

2. Is the manuscript technically sound, and do the data support the conclusions?

Reviewer #1: Yes

Reviewer #2: Yes

3. Has the statistical analysis been performed appropriately and rigorously? 

Reviewer #1: Yes

Reviewer #2: Yes

4. Have the authors made all data underlying the findings in their manuscript fully available?

Reviewer #1: Yes

Reviewer #2: Yes

5. Is the manuscript presented in an intelligible fashion and written in standard English?

Reviewer #1: Yes

Reviewer #2: Yes

6. Review Comments to the Author

Reviewer #1: After reviewing the paper "Effect of lidocaine on salicylate-induced tinnitus in guinea pigs: a focus on the auditory cortex," the author followed all existing comments.

Reviewer #2: (No Response)

7. PLOS authors have the option to publish the peer review history of their article (what does this mean?). If published, this will include your full peer review and any attached files.

Reviewer #1: No

Reviewer #2: No

---

## [Editor Report · Acceptance letter]

27 Jun 2024

PONE-D-24-12496R1 

PLOS ONE

Dear Dr. Ochi, 

I'm pleased to inform you that your manuscript has been deemed suitable for publication in PLOS ONE. Congratulations! Your manuscript is now being handed over to our production team.

Kind regards, 

on behalf of

Dr. Trung Quang Nguyen 

Academic Editor

PLOS ONE